# Characterization and Comparison of Steroidal Glycosides from *Polygonatum* Species by High-Performance Liquid Chromatography–Electrospray Ionization Mass Spectrometry

**DOI:** 10.3390/molecules28020705

**Published:** 2023-01-10

**Authors:** Danyang Liu, Takashi Kikuchi, Wei Li

**Affiliations:** Faculty of Pharmaceutical Sciences, Toho University, Miyama 2-2-1, Funabashi 274-8510, Chiba, Japan

**Keywords:** HPLC-ESI-MS, *Polygonatum* species, steroidal glycosides

## Abstract

*Polygonatum* species have been used as traditional medicines and functional foods in Asia and Europe since ancient times. In this study, a fast and simple method based on liquid chromatography coupled with electrospray ionization mass spectrometry (LC-ESI-MS) was developed to systematically analyze and identify the steroidal glycosides in four major *Polygonatum* species distributed in Japan, including *P. odoratum*, *P. falcatum*, *P. macranthum*, and *P. sibiricum*. As a result, 31 steroidal glycosides were tentatively identified, including 18 known and 13 previously unreported glycosides. Their structures were identified by the interpretation of chromatographic behavior and ESI-MS fragmentation patterns. The identification of 31 steroidal glycosides was indicative of a common biogenetic pathway in *Polygonatum* species. Our study disclosed the chemical profiling of steroidal glycosides in the plants of *Polygonatum* species, which will benefit better phytochemotaxonomical and phytochemical understanding and quality control for their medicinal usage.

## 1. Introduction

The plants belonging to the genus *Polygonatum* (Liliaceae) are perennial herbaceous plants, widely distributed throughout the temperate regions of the Northern Hemisphere, mainly from the Himalayas to Japan [1]. *Polygonatum* species have a long history of being used as traditional medicines and functional foods both in Europe and Asia. In Europe, a *Polygonatum* species with the common name “King Solomon’s-seal” or “Solomon’s seal” is used in folk medicines to treat bruising, rheumatism, and black eye [2]. For Āyurveda in India, *Polygonatum cirrhifolium* Royle and *P. verticillatum* Allioni are imperative ingredients of “Asthaverga”, a group of eight medicinal plants in Ayurveda medicine mostly used as a tonic and aphrodisiac [3].

In China, *Polygonatum* species are an important part of traditional Chinese medicines (TCMs) and have been used since ancient times. The rhizomes of *P. odoratum* (Mill.) Druce are known as “Yuzhu”, which was first documented in Shennong Bencao Jing in about 200 and 250 CE [4]. Meanwhile, the rhizomes of *P. sibiricum* Red., *P. kingianum* Coll. et Hemsl. and *P. cyrtonema* Hua are well-known TCMs named “Huangjing” [4]. Both Yuzhu and Huangjing are traditionally used as a tonic and are included in Chinese Pharmacopoeia (2020) [5], but their medicinal properties have a few differences: Yuzhu (*P. odoratum*) has a particular emphasis on the treatment of lung disease, while Huangjing (*P. sibiricum*, etc.) is more likely to be a health-promoting agent [4].

On the other hand, “Huangjing”, but not “Yuzhu”, is also listed in the Japanese pharmacopoeia 18th edition (JP18) and called “Ohsei” [6]. Ohsei was popular in the Edo period (16th to 18th century) as a tonic and aphrodisiac and is now mainly used as an ingredient in energy drinks. However, because of the variety and similarity of *Polygonatum* species, the original plant of Ohsei (Huangjing) has been not exact for a long time. In fact, the original *P. sibiricum* plant is not a native species of Japan. The common species of *P. falcatum* A. Gray and *P. macranthum* (Maxim.) Koidz. are used as substitutes, as well as *P. odoratum* (Yuzhu in TCMs) [7].

Previous phytochemical investigations revealed that *Polygonatum* species contain a variety of chemical constituents including steroidal glycosides, triterpenoid glycosides, homoisoflavonoids, flavonoids, alkaloids, lignins, phenethyl cinnamides, polysaccharides and lectins [8]. Among them, steroidal glycosides are the most characteristic constituents of *Polygonatum* species [9]. Until now, a total of more than 180 kinds of steroidal glycosides have been isolated from the genus *Polygonatum* [8] and were mainly classified into furostane- and spirostane-type steroidal glycosides. The furostane-type steroidal glycosides are usually 3,26-*O*-bidesmosides with an oligosaccharide chain at C-3 and a D-glucosyl residue at C-26, whereas the spirostane-type glycosides are monodesmosides with an oligosaccharide chain attaches to C-3 [9]. However, the chemical information was almost obtained by the results of individual phytochemical investigations on a single species, and, thus, a parallel comparison of the chemical constituents is desirable.

High-Performance Liquid Chromatography–Electrospray Ionization Mass Spectrometry (HPLC-ESI-MS) has been proven to be a considerably effective analysis tool for the rapid determination of steroidal glycosides in plant extracts with the advantage of high sensitivity and low sample requirement, especially when there is a small sample quantity or no reference compounds are available [10]. In the present study, four *Polygonatum* species (*P. odoratum*, *P. falcatum*, *P. macranthum*, and *P. sibiricum*, which are the most widely distributed in the native flora of Japan, were chosen [11], and their chemical profiles of steroidal glycosides were characterized and compared by a developed HPLC-ESI-MS method.

## 2. Results

### 2.1. Investigation of ESI-MS Fragmentation of Authentic Steroidal Glycosides

Five steroidal glycosides [polygonatumoside F (**4**), timosaponin H1 (**12**), sibiricoside B (**15**), (25*R*)-(3*β*,14*α*)-dihydroxy-spirost-5-ene-3-*O*-*β*-D-glucopyranosyl-(1→2)-[*β*-D-xylopyranosyl-(1→3)]-*β*-D-glucopyranosyl-(1→4)-*β*-D-galacopyranoside (**25**), and 25*S*-aspidistrin (**30**)] were selected as authentic compounds (Figure 1), among which four (**4**, **12**, **25**, and **30**) were obtained in our previous phytochemical investigations [9], and one (**15**) was isolated from *P. odoratum* in the present study. Incidentally, the ^1^H- and ^13^C-NMR spectroscopic data of **15** were fully re-assigned for revising the literature data [12] and given as supporting information in Appendix A. These authentic compounds possess the same sugar chains at C-3 but are different in the aglycone structures. The authentic compounds provided characteristic mass fragment ions and presented regular mass fragmentation pathways, which were useful to reliably identify the steroidal glycosides from *Polygonatum* species.

Polygonatumoside F (**4**) and timosaponin H1 (**12**) were selected for investigating the mass fragmentation pathway of furostane-type steroidal glycosides. The chemical structures of **4** and **12** were different from each other only in the 14-hydroxy moiety. In the negative ion mode, both **4** and **12** provided deprotonated molecule ions [M-H]^−^ as the dominant ion peak. They also provided the fragment ions [M-Xyl-H]^−^ at *m*/*z* 1095 for **4** and *m*/*z* 1079 for **12**, respectively. It is noteworthy that divalent ions peaks [M+2HCOOH-2H]^2−^ and [M+HCOOH-2H]^2−^ were observed at *m*/*z* 659 and 636 for **4**, at *m*/*z* 651 and 628 for **12** (Figure 1a,b), respectively. Conversely, the divalent ions were not detected in spirostane-type steroidal glycosides.

In the positive ion mode, they provided characteristic fragment ions [M-H_2_O+H]^+^ of furostane-type glycosides, which were generated by dehydration of the hydroxy moiety at C-22. Additionally, divalent ions were detected which were observed at *m*/*z* 615 for **4** and at *m*/*z* 607 for **12** as [M+2H]^2+^. Glycosides **4** and **12** possess the same fragmentation pathway for sugar moieties. The fragment ions observed at *m*/*z* 1079 for **4** and at *m*/*z* 1063 for **12** corresponded to the loss of a xylosyl residue (132 Da) as [M-H_2_O-Xyl+H]^+^. The fragment ions at *m*/*z* 755 for **4** and *m*/*z* 739 for **12** corresponded to the sequential loss of two glucosyl residues (162 Da×2) as [M-H_2_O-Xyl-2Glc+H]^+^. They were further fragmented by the loss of a galactosyl residue (162 Da) to generate [M-H_2_O-Xyl-2Glc-Gal+H]^+^ at *m*/*z* 593 and 577, respectively. The fragmentation was in good agreement with the C-3 sugar moiety to be a lycotetraose (**S3**) (Figure 2).

The fragment ions, by further loss of the glucosyl residue at C-26, provided key information to determine the structure of aglycone, particularly the presence of 14-hydroxy moiety. The fragment ions derived from the aglycone were observed at *m*/*z* 413 [Aglycone-2H_2_O+H]^+^ and *m*/*z* 395 [Aglycone-3H_2_O+H]^+^ for **4**, and at *m*/*z* 415 [Aglycone-H_2_O+H]^+^ and *m*/*z* 397 [Aglycone-2H_2_O+H]^+^ for **12**. Consequently, the observation of these fragment ions by the difference of 2 Da between **4** and **12** was useful to identify the hydroxy substituents on the aglycone with one more hydroxy group [9]. It was suggested that dehydration of the hydroxy moiety at the C-14 position and loss of the glucosyl residue at the C-26 position were simultaneously progressed, since the fragment ion peak assigned to [Aglycone-H_2_O+H]^+^ could not have been observed in **4**.

For the spirostane-type steroidal glycosides, sibiricoside B (**15**), (25*R*)-(3*β*,14*α*)-dihydroxy-spirost-5-ene-3-*O*-*β*-D-glucopyranosyl-(1→2)-[β-D-xylopyranosyl-(1→3)]-*β*-D-glucopyranosyl-(1→4)-*β*-D-galacopyranoside (**25**), and 25*S*-aspidistrin (**30**) were selected as authentic compounds. These glycosides are different from each other in C-14 and C-22 hydroxy substitutions.

As shown in Figure 1c–e, spirostane-type steroidal glycosides **15**, **25**, and **30** provided deprotonated molecule ion [M-H]^−^ in the negative ion mode. However, the divalent ions were not detected, which were different from the furostane-type steroidal glycosides. Instead, the monovalent ions [M+HCOOH-H]^−^ were detected. The fragment ions, due to the cleavage of the sugar chain at C-3, were also not detected in the negative ion mode.

In the positive ion mode, the sodium adduct ions [M+Na]^+^ were detected in **15**, **25**, and **30**, but the fragment ion [M-H_2_O+H]^+^ was not detected. The spirostane skeleton contains a spiro-bicyclic acetal at C-22, while the furostane skeleton is usually with a hydroxy moiety at C-22. Thus, observation of the fragment ion [M-H_2_O+H]^+^ corresponding to the dehydration of 22-OH is useful to discriminate between the spirostane- and furostane-type steroidal glycosides.

Further fragment ions were detected, corresponding to the cleavage of the sugar chain at C-3 and dehydration of the hydroxy substitutions in the aglycone. Cleavage of the sugar chain was processed in two steps, firstly, the outer trisaccharide of 2-glucosyl-(3-xylosyl)-glucosyl moiety was lost, and then the inner galactosyl was lost. Meanwhile, dehydration also progressed based on the hydroxy substitutions in the aglycone. As a result, the sequential dominant fragment ions were detected at *m*/*z* 573 [M-Xyl-2Glc-2H_2_O+H]^+^, 411 [Aglycone-2H_2_O+H]^+^, and 393 [Aglycone-3H_2_O+H]^+^ for **15**, at *m*/*z* 575 [M-Xyl-2Glc-H_2_O+H]^+^, 413 [Aglycone-H_2_O+H]^+^, and 395 [Aglycone-2H_2_O+H]^+^ for **25**, and at *m*/*z* 577 [M-Xyl-2Glc+H]^+^, 415 [Aglycone+H]^+^, and 397 [Aglycone-H_2_O+H]^+^ for **30**, respectively.

### 2.2. Identification of Steroidal Glycosides in the Sample Solutions

Four *Polygonatum* plants (*P. odoratum*, *P. falcatum*, *P. macranthum*, and *P. sibiricum*) were pre-treated using a C18 cartridge, and their 80% eluted fractions were analyzed by an LC-ESI-MS operated in a full-scan mode in both positive ion mode and negative ion mode to tentatively identify the steroidal glycosides. The structure was elucidated based on fragment patterns found in authentic steroidal glycosides (see Section 2.1). The total ion chromatograms (TICs) of the sample solutions in positive ion mode are shown in Figure 3. According to the ESI-MS fragmentation patterns, a total of 31 glycosides, including five authentic compounds, were manually detected and designated as **1** to **31** according to the retention times sequence. The molecular fragments detected, retention times, and molecular weight of the 31 glycosides were presented in Table 1. The (+) and (−) ESI-MS and the chemical structures for individual glycosides were given in Appendix A. In the LC-MS analysis of *P. sibiricum*, steroid glycosides were not detected. Since similar results were obtained from *P. sibiricum* harvested in April (data not shown) and July (Figure 3D), this is due to differences in plant metabolism, and not an artifact of sample processing. 

Peaks **1**, **3**, **5**, **6**, **7**, and **8** exhibited fragment ions assignable as [M-H_2_O+H]^+^ as well as two main fragment ions at *m*/*z* 395 and 413, which were the same as peak **4**, suggesting that they are frostane-type glycosides and possess the same aglycone as the authentic compound **4** but differ in the structure of C-3 sugar chains (Appendix A).

Peak **1** exhibited an [M+Na]^+^ ion at *m*/*z* 1281, and an [M-H]^−^ ion at *m*/*z* 1257, suggesting the molecular weight of **1** was 30 Da more than that of **4** (Appendix A). Since **1** and **4** produced the same fragment ions at *m*/*z* 755, 593, 575, and 557 by cleavage of the C-3 sugar chain, the difference in molecular weight was most like due to the replacement of the terminal xylose moiety in **4** with glucose in **1**. The sugar chain of commetetraose (**S1**) in **1** is common in known steroidal glycosides from *Polygonatum* plants (Figure 2) [8]. Thus, **1** was identified to be 26-*O*-(*β*-D-glucopyranosyl)-furost-5-ene-3*β*,14*α*,22*α*,26-tetraol 3-*O*-*β*-D-glucopyranosyl-(1→2)-[*β*-D-glucopyranosyl-(1→3)]-*β*-D-glucopyranosyl-(1→4)-*β*-D-galactopyranoside.

Peak **3** exhibited fragment ions (*m*/*z* 917, 755, and 593) generated by sequential losses of three hexose residues (Appendix A). Based on comparison with the data of **4**, the C-3 sugar chain of **3** was most like to be a trisaccharide of *β*-D-glucopyranosyl-(1→2)-*β*-D-glucopyranosyl-(1→4)-*β*-D-galactopyranose (**S2**), which lost a xylosyl moiety from **S1** (Figure 2). Thus, **3** was identified to be 26-*O*-(*β*-D-glucopyranosyl)-furost-5-ene-3*β*,14*α*,22*α*,26-tetraol 3-*O*-*β*-D-glucopyranosyl-(1→2)-*β*-D-glucopyranosyl-(1→4)-*β*-D-galactopyranoside, which was previously isolated from *P. odoratum* [13].

As for peaks **5**, **7**, and **8**, two fragment ions were observed at *m*/*z* 797 and 593, with a 204 Da (162 Da + 42 Da) mass difference, suggesting the inner galactose was acetylated (Appendix A). Taking account the known steroidal glycosides from *Polygonatum* plants [14], peak **5** was identified to be 26-*O*-(*β*-D-glucopyranosyl)-furost-5-ene-3*β*,14*α*,22*α*,26-tetraol 3-*O*-*β*-D-glucopyranosyl-(1→2)-[*β*-D-glucopyranosyl-(1→3)]-*β*-D-glucopyranosyl-(1→4)-2″-*O*-acetyl-*β*-D-galactopyranoside.

Both **7** and **8** were also considered to have galactosyl residue with an acetyl group in the C-3 sugar chain such as **5**. The ions at *m*/*z* 959 and 797 from **7** were formed from the sequential losses of two hexose residues from the [M-H_2_O+H]^+^ ion at *m*/*z* 1121. Therefore, **7** was identified to be 26-*O*-(*β*-D-glucopyranosyl)-furost-5-ene-3*β*,14*α*,22*α*,26-tetraol 3-*O*-*β*-D-glucopyranosyl-(1→2)-*β*-D-glucopyranosyl-(1→4)-2″-*O*-acetyl-*β*-D-galactopyranoside, a compound that does not have a terminal glucose moiety of sugar chain. Meanwhile, the ion at *m*/*z* 797 from **8** was formed from the losses of two hexoses and one pentose residues (456 Da) from the [M-H_2_O+H]^+^ ion at *m*/*z* 1253. According to the saponin with a *β*-D-glucopyranosyl-(1→2)-[*β*-D-xylopyranosyl-(1→3)]-*β*-D-glucopyranosyl-(1→4)-2″-*O*-acetyl-*β*-D-galactopyranose moiety (**S5**) at C-3 position having been isolated from *P. odoratum* [15], **8** was identified as shown in Figure 2.

In peak **6** MS, the ion at *m*/*z* 593 was 146 Da different than the ion at *m*/*z* 739 [M-H_2_O-Xyl-2Glc+H]^+^, which corresponded to the loss of a deoxyhexose residue (146 Da) (Appendix A). The rhamnose is the usual constituent sugar in the genus *Polygonatum* [8]. Thus, **6** was determined to be 26-*O*-(*β*-D-glucopyranosyl)-furost-5-ene-3*β*,14*α*,22*α*,26-tetraol 3-*O*-*β*-D-glucopyranosyl-(1→2)-[*β*-D-xylopyranosyl-(1→3)]-*β*-D-glucopyranosyl-(1→4)-*α*-L-rhamnopyranoside.

Peak **13** exhibited the fragment ions of *m*/*z* 901 [M-2Rha+H]^+^, 755 [M-3Rha+H]^+^, and 575 [M-Glc-3Rha-H_2_O+H]^+^, which were formed from the loss of three deoxyhexose and one pentose moieties (Appendix A). In addition, the fragment ions of *m*/*z* 413 [Aglycone-H_2_O+H]^+^ and 395 [Aglycone-2H_2_O+H]^+^ suggested a glucose residue at the C-26 position and the same aglycone as **4**. The sugar moiety (**S8**) of **13** at the C-3 position was reported as those of parrisaponin Pb from *P. kingianum* [16] and *P. zanlanscianense* [17], and **13** was identified as shown in Figure 2.

Peak **2** obtained the ions at *m*/*z* 753 and 591 by the sequential losses of the sugar residues in the C-3 sugar chain of **S3** (Figure 2) from the [M-H_2_O+H]^+^ ion (*m*/*z* 1209) (Appendix A). The ion at *m*/*z* 411 from **2** was formed from the loss of the glycosyl residue at C-26, and then the elimination of the hydroxyl groups yielded the ion at *m*/*z* 393 [Aglycone-2H_2_O+H]^+^ and *m*/*z* 375 [Aglycone-3H_2_O+H]^+^. These suggested that **2** has two free hydroxyl groups and a double bond on the aglycone, since each of the above fragment ions has a 2 Da mass difference from **4**. The double bond was conjectured to be located at C-25 and C-27, since such furostane-type steroids have been reported in Liliaceae plants [18,19]. Thus, **2** was determined to be 26-*O*-(*β*-D-glucopyranosyl)-furost-5,25(27)-diene-3*β*,14*α*,22*α*,26-tetraol 3-*O*-*β*-D-glucopyranosyl-(1→2)-[*β*-D-xylopyranosyl-(1→3)]-*β*-D-glucopyranosyl-(1→4)-*β*-D-galactopyranoside.

Peaks **17** and **18** provided the same fragment ions and confirmed a pair of isomers. The ions of [M-H_2_O+H]^+^ at *m*/*z* 593 and [Aglycone-H_2_O+H]^+^ at *m*/*z* 431 suggested the molecular weight to be 610 Da, and **17** and **18** having a glucosyl moiety at C-26 position as seen in other *Polygonatum* saponins (Appendix A). By comparing retention times and mass spectra with standard peaks, **17** and **18** could be unequivocally identified as 25*R*- and 25*S*-polygonatumoside G, respectively, which have been reported in *P. odoratum* [9]. Peak **16** showed a similar mass fragmentation pattern with **17** and **18**, except in all mass fragment ions have a 2 Da mass difference, which suggested the existence of a double bond in the structure (Appendix A). Thus, **16** was identified as furost-5,25(27)-diene-1*β*,14*α*,22*α*,26-tetraol 26-*O*-*β*-D-glucopyranoside, the sapogenin of **2** (**A2**) (Figure 2).

Peak **10** exhibited fragment ions that were less than 2 Da than those of the authentic compound **12** (Appendix A). Then, the ions at *m*/*z* 413 and 395 were characterized by the loss of a glucosyl residue at the C-26 position and H_2_O at the C-3 position and determined the aglycone skeleton with one more double bond than **12**. The double bond was conjectured to be located at C-25 and C-27 like **2** and **16**. Thus, **10** was inferred as 26-*O*-(*β*-D-glucopyranosyl)-furost-5,25(27)-diene-3*β*,22*α*,26-triol 3-*O*-*β*-D-glucopyranosyl-(1→2)-[*β*-D-xylopyranosyl-(1→3)]-*β*-D-glucopyranosyl-(1→4)-*β*-D-galactopyranoside.

Peaks **20** and **21** followed the same fragmentation pathway with **16**, except all mass fragment ions have 16 Da and 18 Da mass differences, respectively (Appendix A). Thus, **20** was considered as a dehydroxy product of **16**, and could be assigned as furost-5,25(27)-diene-3*β*,22*α*,26-triol 26-*O*-*β*-D-glucopyranoside, which has the same aglycone as **10**. Meanwhile, **21** was considered as a reduction product of **20**, and could be identified as funkioside B (**21**) [20], since the fragment ions of **21** have more 2 Da than those of peak **20**.

Peaks **9**, **11**, and **14** followed similar fragmentation patterns to the authentic compound **12**, and all exhibited the main fragment ions at *m*/*z* 577, 415 [Aglycone-H_2_O+H]^+^ and 397 [Aglycone-2H_2_O+H]^+^ (Appendix A), which suggested that they have the same aglycone as **12** possessing the glucosyl moiety at C-26 position but with differences in C-3 sugar chains. Peak **9** exhibited the fragment ions at *m*/*z* 901, 739, and 577, which were formed from the sequential losses of four hexose moieties from the [M-H_2_O+H]^+^ ion at *m*/*z* 1225, and was conjectured as polyfuroside, which was previously isolated from *P. officinale* [21], *Solanum nigrum L* [22] and *Allium macrostemon* [23] with the C-3 sugar chain of **S1** (Figure 2). Peak **11** showed the same fragmentation pattern as **3** with the sugar chain of **S2** (Figure 2), and identified as 26-*O*-(*β*-D-glucopyranosyl)-furost-5-ene-3*β*,22*α*,26-triol 3-*O*-*β*-D-glucopyranosyl-(1→2)-*β*-D-glucopyranosyl-(1→4)-*β*-D-galactopyranoside. Peak **14** exhibited a similar fragment pattern to **5**. Therefore, **14** was identified as 26-*O*-(*β*-D-glucopyranosyl)-furost-5-ene-3*β*,22*α*,26-triol 3-*O*-*β*-D-glucopyranosyl-(1→2)-[*β*-D-glucopyranosyl-(1→3)]-*β*-D-glucopyranosyl-(1→4)-2″-*O*-acetyl-*β*-D-galactopyranoside.

Peak **19** presented the fragment ions at *m*/*z* 739, 577, 415, and 395 same with **12**, but the [M-H_2_O+H]^+^ ion did not exist, suggesting that there is no hydroxyl group connected to the C-22 position (Appendix A). The ions at *m*/*z* 739 and 577 were formed from the sequential losses of C-3 sugar chains as **S3** (Figure 2) from the ion at *m*/*z* 1217. Thus, **19** was identified to be polygodoside G, which was previously isolated from *P. odoratum* [15].

On the other hand, 11 peaks were identified as spirostane-type steroidal glycosides by no detection of the [M-H_2_O+H]^+^ and formation of the [M+HCOOH-H]^−^ ions in high abundance.

Peaks **24**, **25**, **26**, and **27** exhibited the same fragment ions at *m*/*z* 431, 413, and 395, suggesting that they have the same aglycone(Appendix A). In addition, a fragment ion at *m*/*z* 575 suggested that the internal galactose in the sugar chain was also common.

Peak **26** was considered to be an isomeric compound of the authentic compound **25** with the same molecular weight of 1049 Da. Peak **26** could be inferred as the stereoisomer of **25** at the C-25 position since almost the same fragment ions were observed in **25**.

Peaks **24** and **27** exhibited the almost same fragmentation patterns (Appendix A) as **25** and **26**, suggesting that they have the same aglycone skeleton but differ in the sugar moieties. Peak **24** has a molecular weight of 1078 Da, and its fragment ions at *m*/*z* 575 and 413 were formed by the sequential losses of sugar residues from **S1** (Figure 2). Peak **27** has a molecular weight of 916 Da, differing 132 Da corresponding to a pentose residue from **25**, therefore having the same C-3 sugar chains as **3** (**S2**) (Figure 2). Thus, peak **24** was identified as pod Ⅲ [15] and **27** was identified as polygonatumoside D [24]. Both compounds have been previously reported in *P. odoratum*.

Both **22** and **23** observed the ions at *m*/*z* 573, 411, and 393, suggesting that they have the same aglycone skeleton but different sugar moieties (Appendix A). Peak **23** exhibited the [M+Na]^+^ ion less 2 Da than authentic **25** (**23**: *m*/*z* 1069; **25**: *m*/*z* 1071) but the same fragment ions at *m*/*z* 299 (**ii**), 281 (**iii**), 269 (**iv**), and 251 (**v**), which assigned the characteristic fragments of the steroidal nucleus [25], were observed (Figure 4). This suggested that the aglycones of **23** (*m*/*z* 429 [Aglycone+H]^+^) and **25** (*m*/*z* 431 [Aglycone+H]^+^ (**i**)) have the same A, B, C, D, and E rings and sugar chain, but peak **23** has a carbon-carbon double bond in the F ring [15]. Peak **22** presented an [M+Na]^+^ ion at *m*/*z* 1099 and has a molecular weight of more than 30 Da than **23**. The C-3 sugar chain was suggested as **S1** (Figure 2). Thus, **23** and **22** were identified as polygodoside A and polygodoside B, respectively [15]. Both compounds have been previously reported in *P. odoratum*.

Peak **28** produced similar fragment ions to authentic compound **30**, except for the 2 Da mass difference of all fragment ions, which suggested that **28** has one more double bond than **30** (Appendix A). Thus, **28** was identified as 25(*S*)-spirost-5-ene-3*β*-ol 3-*O*-*β*-D-glucopyranosyl-(1→2)-[*β*-D-xylopyranosyl-(1→3)]-*β*-D-glucopyranosyl-(1→4)-galactopyranoside, which has been previously reported in *P. odoratum* [26].

Peaks **29** and **31** exhibited the fragment ions at *m*/*z* 577, 415, and 397 (Appendix A), suggesting that they have the same aglycone skeleton as **30** but differ in sugar moieties. Peaks **29** and **31** have a 30 Da (for -CH_2_O-) and 132 Da (for a pentose moiety) mass difference from **30**, respectively. Thus, peak **29** could be assigned as pod Ⅳ, which has been previously reported in *P. odoratum* [15], and **31** as neosibiricoside D, which has been previously reported in both *P. odoratum* [14] and *P. sibiricum* [27].

In general, the retention time of frostane-type steroids was shorter (Rt 8.90–21.08 min) than those of spirostane-types (Rt 22.29 min–31.16 min), except for the spirostane steroid **15** (Appendix A) having the hydroxy group in the F-ring (Rt 15.14 min). The retention times, depending on the types of the sugar chain, can be summarized as follows. In the case of the frostane-type steroids, the retention time of compounds having the sugar chain **S1** (-Gal(-Glc)-Glc-Glc) is shorter than that of **S2** (-Gal(-Glc)-Glc). Meanwhile, the steroids having the sugar chain **S3** (-Gal(-Glc)-Glc-Xyl) had a longer retention time than that of **S2** (Rt **1** < **3** < **4**; **9** < **11** < **12**). This retention time relationship is the same as the steroids possessing sugar chains of **S5**, **S6**, and **S7** which are acetylated at 2-OH of the inner galactose (Rt **5** < **7** < **8**). In addition, a steroid in which galactose in the **S3** sugar chain of steroidal saponin was replaced with rhamnose (**S7**) had a longer retention time (Rt **4** < **6**). On the other hand, among spirostane glycosides, the one with **S1** had the shortest retention time, but the one with **S3** had a shorter retention time than the one with **S2**. (Rt **24** < **25** < **26** < **27**; **29** < **30** < **31**).

### 2.3. The Biogenetic Pathway of Steroidal Glycosides in Polygonatum Species

As shown in Figure 3, a total of 15 steroidal glycosides were found in *P. odoratum*, 22 in *P. falcutum*, and 23 in *P. macranthum*, but none in *P. sibiricum*. All varieties collected in October showed fewer species of steroidal glycosides. On the contrary, the plants collected in July showed the most abundant steroidal glycosides among all the varieties. A possible biogenesis pathway is shown in Figure 5, according to the analysis result of plants collected in July.

Steroidal glycosides in *Polygonatum* species could be divided into spirostane- and furostane-types base on the aglycone structures (Appendix A). Spirostane-types are always the monodesmosides with a sugar chain attached to the C-3 position, and the furostane- types are always the bisdesmosides with an additional glucopyranosyl moiety linked in the C-26 position. Our previous studies about steroidal glycosides [9] and cholestane-type glycosides [24] reported that steroidal glycosides in genus *Polygonatum* have the same common biosynthetic precursor, (22*S*)-cholest-5-ene-3*β*,16*β*,22-triol. C-22 oxidation of the precursor produced 16,26-dihydroxy-22-keto-cholesterol, then the biosynthesis routes were divided into two classes. One group is characterized by 14-hydroxylation, followed by 3,26-glycosylation, which produces the aglycone polygonatumoside (**A2**) (Appendix A), while the other group is not, and produces the aglycone **A4**, a furostane-type aglycone without 14-OH. The furostane-type aglycones **A2** and **A4** deglycosylated at C-26, followed by a ring closure to 26-OH take place with dehydration of 22-OH, producing the spirostane-type aglycone **A8** and **A10**, respectively. The furostane- and spirostane-type aglycones undergo various modifications in a series of biosynthetic reactions, including hydroxylation and dihydroxylation, to enable further diversification.

The original plants of Huangjing (*P. falcutum* and *P. macranthum*) showed a more complex oxidation process than the original plant of Yuzhu (*P. odoratum*) at the aglycone (Figure 5). Significantly, the original plants of Huangjing (*P. falcutum* and *P. macranthum*) have the acetyl groups on hydroxyl groups at the sugar residues (**S4**, **S5**, and **S6**) which were different from Yuzhu (*P. odoratum*). On the other hand, the steroidal glycosides from *Polygonatum* species mainly have the sugar moiety as lycotetraose (**S3**). The main components **4**, **12**, **15**, **25**, **26**, and **30**, which were detected in all varieties with high abundance, all have the C-3 sugar chain as **S3**.

The results showed a similar chemical composition of steroidal glycosides among the *Polygonatum* species cultivated in Japan, but it also has some interspecific differences between the two groups of Huangjing and Yuzhu. On the whole, *P. falcatum* and *P. macranthum* contained steroidal glycosides with higher oxidation levels, while the sugar moiety also with structural differences from *P. odoratum*.

## 3. Materials and Methods

### 3.1. General Methods

Acetonitrile (MeCN), formic acid (HCOOH), and water (H_2_O) were LC/MS-grade and obtained from Fujifilm Wako Pure Chemical Industries, Ltd. (Osaka, Japan). The ^1^H and ^13^C NMR spectra were measured on a JEOL ECA-500 spectrometer (Tokyo, Japan) with the measuring deuterated solvent as the internal reference, and the chemical shifts were expressed in ppm units. Column chromatography was performed on Diaion HP-20 (Mitsubishi Chemical Corporation, Tokyo, Japan) and ODS (100–200 mesh, Chromatorex DM1020T ODS, Fuji Silysia Chemical Co., Ltd., Aichi, Japan). Semi-preparative HPLC was performed on a Waters 600E HPLC pump (Waters Corp., Milford, MA, USA) with a Shimadzu SPD-10A (Kyoto, Japan) intelligent ultraviolet/visible (UV/vis) detector, a Shodex RI-72 differential refractometer detector (Shoko Science Co., Ltd., Tokyo, Japan), and an RP-C18 silica gel column (YMC Actus Triart C18, 150 × 20 mm I.D., Kyoto, Japan).

### 3.2. Plant Materials

Plant materials were shown in Figure 5 and Table 2. The rhizomes of *Polygonatum* species were cultivated in Toho University Botanical Garden of Medicinal Plants, Japan, and were collected from April to October 2018. All of the plants were identified by one of the authors, WL. The voucher specimens were deposited in the Department of Pharmacognosy, Faculty of Pharmaceutical Sciences, Toho University.

### 3.3. Preparation of the Sample Solutions

The dried rhizomes were pulverized into homogeneous powders. The powder (0.5 g) was ultrasonically extracted with methanol (MeOH, 10 mL) at room temperature for 1 h. After evaporation, the extracts were ultrasonically dissolved in MeOH (1.5 mL) and H_2_O (3.5 mL) was added to make a 30% MeOH solution (5 mL). The solution was loaded onto a Sep-Pak C18 plus short cartridge (Waters), eluted with 30% MeOH/H_2_O solution (*v*/*v*) (10 mL), and 80% MeOH/H_2_O solution (*v*/*v*) (10 mL). The 80% MeOH/H_2_O elution (5 mL) was dried in vacuo and then dissolved in 1 mL MeOH. The extracted solutions were filtered through a 0.45 µm syringe filter and used as sample solutions.

### 3.4. LC-MS Analysis

LC-MS analysis was performed using an LCMS-8040 triple quadrupole LC/MS/MS mass spectrometer (Shimadzu Co., Ltd., Kyoto, Japan). HPLC was performed on a YMC-Triart C18 column (3.0 μm, 150 × 20 mm I.D.) maintained at 35 °C. The flow rate was 0.2 mL/min. The mobile phase was composed of A (0.1% HCOOH in H_2_O) and B (0.1% HCOOH in MeCN) with a gradient elution: 0–40 min, 20–70% B; 40–50 min, 100% B. The column was equilibrated for 15 min under the initial conditions. The injection volume was 4 µL for the qualitative analysis.

The mass spectrometer was operated in both positive- and negative-ion modes in the range of *m*/*z* 100 to 1600. The ESI parameters were as follows: interface voltage, 4.5 kV in the positive-ion mode and −3.5 kV in the negative-ion mode; dry gas, 15 L/min; and dry temperature, 350 °C.

### 3.5. Isolation of Sibiricoside B (***15***)

The air-dried rhizomes of *P. odoratum* (0.9 kg) were cut into pieces and were ultrasonically extracted with MeOH (2 L) for 1 h at room temperature. After removal of the solvent by evaporation, the extract (138 g) was subjected to Diaion HP-20 column chromatography and eluted with gradients of H_2_O and 40%, 80%, and 100% MeOH. The 80% MeOH eluate was chromatographed on semi-preparative reversed-phase (RP)-HPLC to afford sibiricoside B (**15**) (14 mg).

## 4. Conclusions

In this study, the steroidal glycosides in 4 species of genus *Polygonatum* were systematically analyzed and identified by a simple, rapid, and sensitive LC-ESI–MS method. The mass fragmentation pattern of spirostane-type and furostane-type steroidal glycosides was also investigated, and 31 compounds were tentatively identified. Our study provided chemical support for further phytochemotaxonomical studies of *Polygonatum* species and provided evidence for the identification of steroidal glycosides by an LC-ESI–MS method when no reference compounds were available.

## Figures and Tables

**Figure 1 molecules-28-00705-f001:**
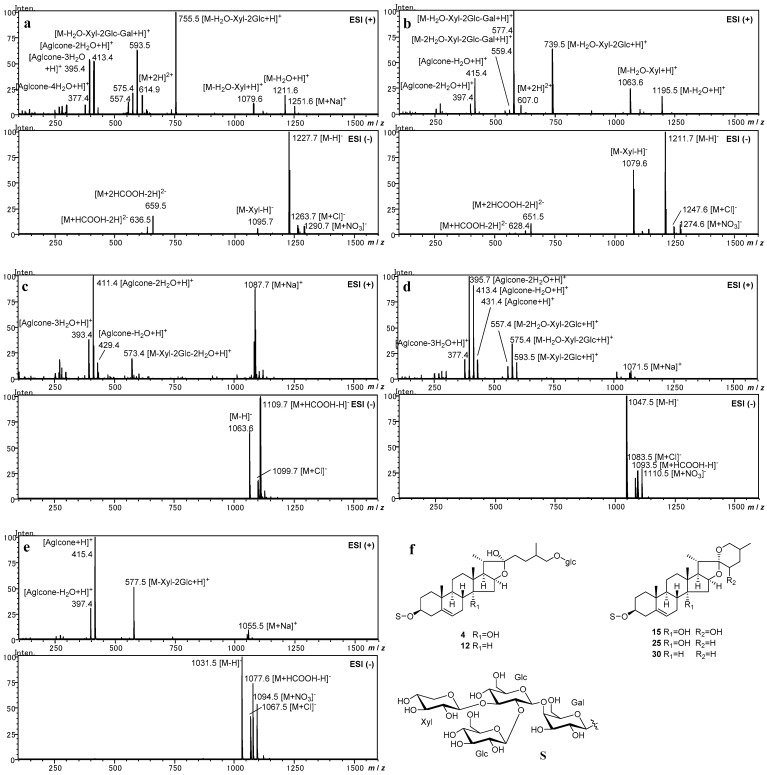
ESI-MS spectra and structure of authentic steroidal glycosides: (**a**) ESI-MS spectra of polygonatumoside F (**4**); (**b**) ESI-MS spectra of timosaponin H1 (**12**); (**c**) ESI-MS spectra of sibiricoside B (**15**); (**d**) ESI-MS spectra of (25*R*)-(3*β*,14*α*)-dihydroxy-spirost-5-ene-3-O-*β*-D-glucopyranosyl-(1→2)-[*β*-D-xylopyranosyl-(1→3)]-*β*-D-glucopyranosyl-(1→4)-*β*-D-galacopyranoside (**25**); (**e**) ESI-MS spectra of 25*S*-aspidistrin (**30**); (**f**) Structure of authentic steroidal glycosides.

**Figure 2 molecules-28-00705-f002:**
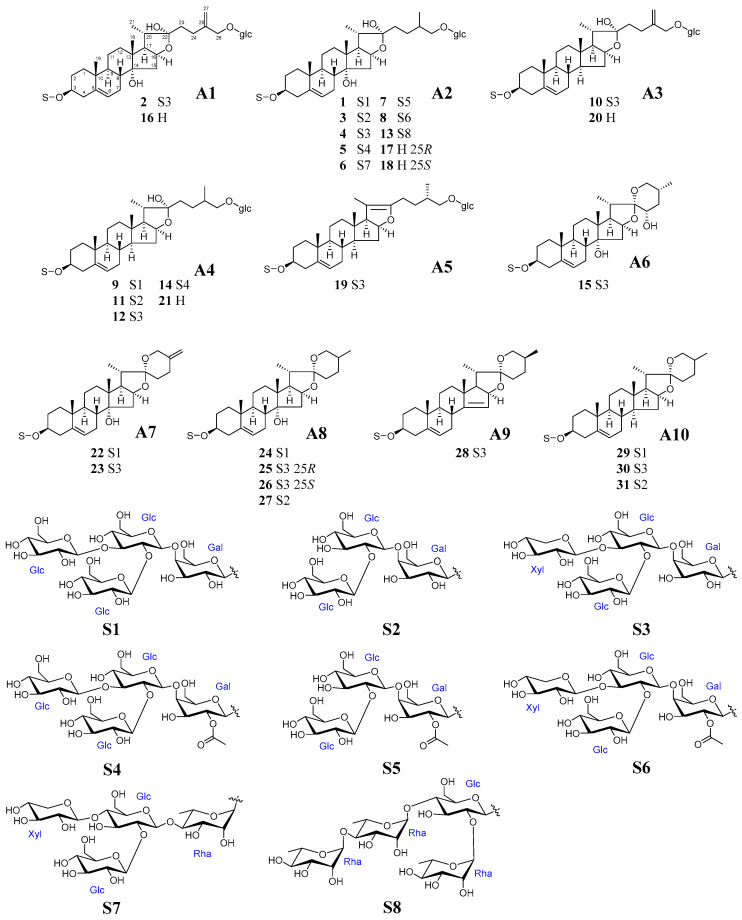
Chemical structures of the steroidal glycosides in *Polygonatum* species.

**Figure 3 molecules-28-00705-f003:**
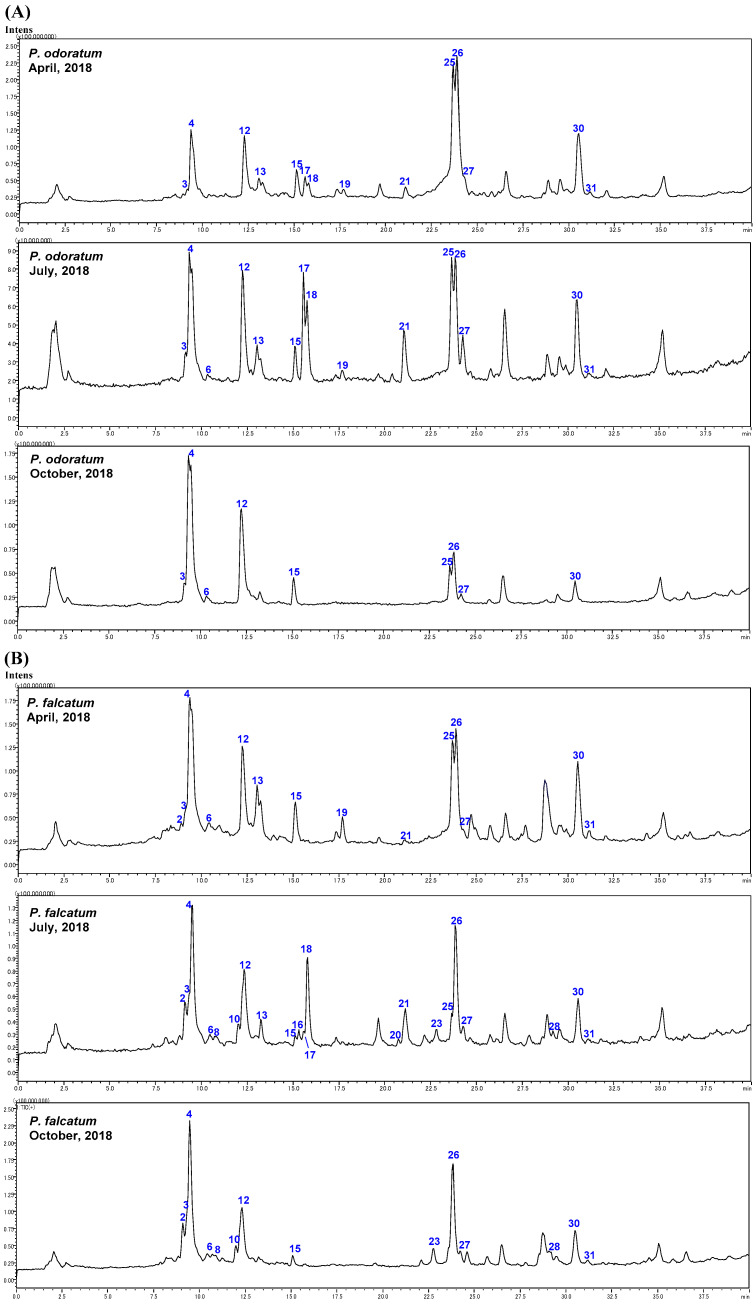
Total ion chromatograms of the extracts from the rhizomes of four *Polygonatum* species in the positive-ionization mode. (**A**) *P. odoratum*. (**B**) *P. falcatum*. (**C**) *P. macranthum*. (**D**) *P. sibiricum*.

**Figure 4 molecules-28-00705-f004:**
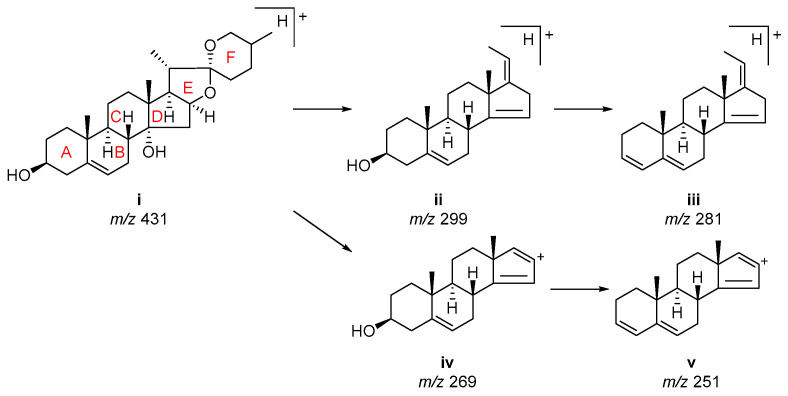
Proposed fragmentation pathway for the aglycone of compound **25**.

**Figure 5 molecules-28-00705-f005:**
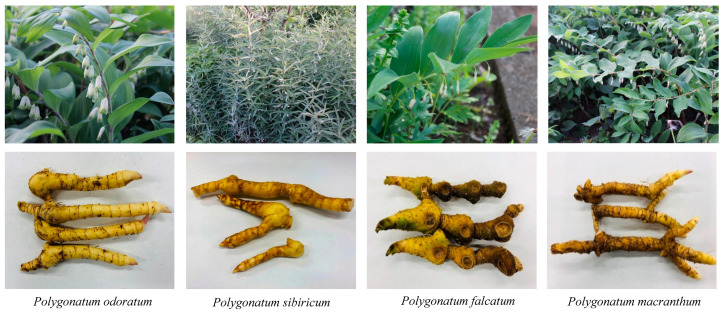
Photographs of plants from genus *Polygonatum*.

**Table 1 molecules-28-00705-t001:** Steroidal saponins detected from *Polygonatum* plants by LC-ESI-MS analysis.

Peaks	Structure		R_t_ (min)	Formula	MW	Positive Ion (*m*/*z*)									Negative Ion (*m*/*z*)			Occurrence
	Aglycone	Sugar				[M+Na]^+^	[M-H_2_O+H]^+^	[Aglycone-H_2_O+H]^+^	[Aglycone- 2H_2_O+H]^+^	[Aglycone- 3H_2_O+H]^+^	Others				[M-H]^−^	[M+Cl]^−^	[M+HCOOH-H]^−^	
1	A2	S1	8.90	C_57_H_94_O_30_	1259.4	1281.6	1241.6		413.4	395.4	755.5	575.4			1257.7	1293.7		PM
2	A1	S3	9.11	C_56_H_90_O_29_	1227.3	1249.6	1209.6		411.4	393.4	753.5	591.4			1225.7	1261.7		PF, PM
3 ^b^	A2	S2	9.18	C_51_H_84_O_25_	1097.2	1119.6	1079.6		413.5	395.3	917.6	755.5	593.5		1095.5			PO, PF, PM
4 ^a,b^	A2	S3	9.38	C_56_H_92_O_29_	1229.3	1251.6	1211.6		413.3	395.4	755.5	593.5	575.4	557.4	1227.7	1263.7		PO, PF, PM
5	A2	S4	10.28	C_59_H_96_O_31_	1301.4	1323.6	1283.6		413.4	395.4	797.5	593.5			1299.7	1335.7		PM
6	A2	S7	10.31	C_56_H_92_O_28_	1213.3	1235.6	1195.6		413.4	395.5	739.5	593.5			1211.7	1247.5		PO, PF
7	A2	S5	10.66	C_53_H_86_O_26_	1139.2	1161.6	1121.6		413.7	395.6	959.5	797.7	593.5		1137.7	1173.8		PM
8	A2	S6	10.78	C_58_H_94_O_30_	1271.4	1293.6	1253.5		413.4	395.4	797.6	755.4	593.6		1269.8	1305.7		PF
9 ^b^	A4	S1	11.87	C_57_H_94_O_29_	1243.4	1265.6	1225.6	415.4	397.4		901.6	739.5	577.4		1241.7	1277.7		PM
10	A3	S3	12.01	C_56_H_90_O_28_	1211.3	1233.6	1193.6	413.3	395.3		1031.5	899.4	737.6	575.4	1209.7	1245.5		PF, PM
11	A4	S2	12.18	C_51_H_84_O_24_	1081.2	1103.6	1063.6	415.5	397.5		901.6	739.6	577.4		1079.6	1115.6		PM
12 ^a,b^	A4	S3	12.28	C_56_H_92_O_28_	1213.3	1235.6	1195.7	415.4	397.4		1033.6	901.6	739.6	577.5	1211.7	1247.7		PO, PF, PM
13	A2	S8	13.08	C_57_H_94_O_27_	1211.4	1233.6			413.4	395.4	901.6	755.4	575.5		1209.7	1245.6	1255.7	PO, PF
14	A4	S4	13.32	C_59_H_96_O_30_	1285.4	1307.6	1267.6	415.5	397.4		943.7	781.5	577.4		1283.7	1319.7		PM
15 ^a,b^	A6	S3	15.14	C_50_H_80_O_24_	1065.2	1087.6			411.4	393.4	573.4				1063.6	1109.6		PO, PF, PM
16	A1	H	15.32	C_33_H_52_O_10_	608.8	631.4	591.5								607.4		653.4	PF
17 ^b^	A2	H	15.59	C_33_H_54_O_10_	610.8	633.5	593.5	431.4							609.4		655.5	PO, PF, PM
18 ^b^	A2	H	15.78	C_33_H_54_O_10_	610.8	633.4	593.5	431.2							609.4		655.5	PO, PF, PM
19 ^b^	A5	S3	17.70	C_56_H_90_O_27_	1195.3	1217.7		397.4			739.5	577.5	415.4		1193.7	1229.8	1239.8	PO, PF
20	A3	H	20.74	C_33_H_52_O_9_	592.8	615.4	575.4		413.5								637.5	PF
21 ^b^	A4	H	21.08	C_33_H_54_O_9_	594.8	617.5	577.5	415.3									639.5	PO, PF, PM
22 ^b^	A7	S1	22.29	C_51_H_80_O_24_	1077.2	1099.6		411.4	393.4		573.6				1075.6		1121.6	PM
23 ^b^	A7	S3	22.81	C_50_H_78_O_23_	1047.2	1069.6		411.4	393.3		573.4				1045.6		1091.6	PF, PM
24 ^b^	A8	S1	23.12	C_51_H_82_O_24_	1079.2	1101.6		413.4	395.4		575.4	431.4			1077.7		1123.6	PM
25 ^a,b^	A8	S3	23.69	C_50_H_80_O_23_	1049.2	1071.6		413.4	395.7		575.2	557.2	431.4		1047.6		1093.6	PO, PF, PM
26 ^b^	A8	S3	23.89	C_50_H_80_O_23_	1049.2	1071.5		413.4	395.4		575.4	557.4	431.4		1047.6		1093.6	PO, PF, PM
27 ^b^	A8	S2	24.26	C_45_H_72_O_19_	917.1	939.5		413.4	395.4		575.4	431.4			915.5		961.6	PO, PF, PM
28 ^b^	A9	S3	29.21	C_50_H_78_O_22_	1031.2	1053.6		395.4			575.4	413.4			1029.6		1075.6	PF
29 ^b^	A10	S1	29.71	C_51_H_82_O_23_	1063.2	1085.6		397.3			577.4	415.3			1061.5		1107.7	PM
30 ^a,b^	A10	S3	30.52	C_50_H_80_O_22_	1033.2	1055.6		397.4			577.5	415.4			1031.6		1077.6	PO, PF, PM
31 ^b^	A10	S2	31.16	C_45_H_72_O_18_	901.1	923.6		397.4			577.7	415.4			899.6		945.6	PO, PF, PM

^a^ Structurally confirmed by comparison with reference compound. ^b^ Reference compound.

**Table 2 molecules-28-00705-t002:** Plant materials.

No.	Species	Sample	Medicinal Parts
1	*P. odoratum*	PO4	Rhizomes
2	*P. odoratum*	PO7	Rhizomes
3	*P. odoratum*	PO10	Rhizomes
4	*P. falcatum*	PF4	Rhizomes
5	*P. falcatum*	PF7	Rhizomes
6	*P. falcatum*	PF10	Rhizomes
7	*P. macranthum*	PM7	Rhizomes
8	*P. macranthum*	PM10	Rhizomes
9	*P. macranthum*	PM10-2	Rhizomes
10	*P. sibiricum*	OS7	Rhizomes

## Data Availability

The data presented in this study and samples of the compounds are available upon request from the corresponding author.

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
