# Peer review of "Characterization and Comparison of Steroidal Glycosides from Polygonatum Species by High-Performance Liquid Chromatography–Electrospray Ionization Mass Spectrometry"

_molecules, 2023, doi:10.3390/molecules28020705_

Round 1

Reviewer 1 Report

Please check the attached file

Reviewer 2 Report

This is a very interesting paper reporting the utility of LC-MS to characterize glycosides present in various members of the Polygonatum genus. The authors used some standard glycosides and examined the fragmentation patterns of the glycosides extracted from the plants to establish structures, including some previously unreported glycosides. Their methodology is extremely sound and as a mass spectrometrist who performs structural analysis frequently, the reviewer is very impressed with their interpretation skills and their logic in assigning the proposed structures (for example, when they have concluded--I am certain correctly--of acetylation features). This paper should be published, and I only have few minor comments.

1. lines 142 and 145. Here the authors use "aglycon" when I think they meant "aglycone".

2. line 164. Use the word "designated" as opposed to "design" here.

3. Figure 3d and line 168. It is interesting and a little surprising that considering the success in detecting glycosides in the other specimens that none were detected in P. sibiricum. Was this the case over multiple harvesting times? If so, it indicates a real difference in the plant's metabolism and not due to an artifact of sample processing. It would be good to comment on whether this observation held at different times of harvest.

Reviewer 3 Report

The manuscript by Liu with co-workers is dedicated to an identification and structural studies of a number of steroidal glycosides in medicinal plant extracts by LC-ESI-MS. It is well written and makes a good impression. The authors very carefully analyzed the complex mass spectra of the analytes which allowed them to discriminate even close structures in real samples of plant extracts. However, there are some issues that need to be clarified or corrected prior recommending the manuscript for the publication:

1.     In my opinion, the introduction is too long in describing the medical use of the plant to the detriment of information about the current state of research on steroidal glycosides in its composition.

2.     For the identification, the authors use only low-resolution mass spectra, without involving tandem mass spectrometry and without using standards.  Thus, the use of the term “tentative identification” seems to be more correct in this study.

3.     In Abstract section, the authors declare the development of the fast and simple method to analyze and identify steroidal glycosides in four major Polygonatum species. Unfortunately, further in the text this issue is not fully disclosed. In the Results section, it would be useful to provide a brief summary of the developed method and the algorithm for its application.

4.     It is surprising that the authors do not use tandem mass spectrometry, which would increase the reliability of analyte identification by obtaining deeper structural information. Moreover, in the analysis of real samples, the peaks of different steroidal glycosides can overlap making an interpretation of mass spectra too difficult or even erraneous.

5.     In Experimental section the authors mention that collision voltage of 15V was used. What does it mean? Does it relate to ion optics (QArray) or collision induced fragmentation of all analytes in a collision cell? This parameter can be crucial in obtaining informative MS data and thus should be optimized.

Round 2

Reviewer 1 Report

The LC-MS methodology used in the study can not produce reliable results for the identification of compounds. Unfortunatelly the authors do not have the ability to carry out the proposed major revisions, which could produce acurate and reliable data. 

Author Response

We understand the major concern from the reviewer may be that in this study, a high-resolution mass spectrometer, which affords high mass accuracy measurements, could not have been used. In this study, we successfully achieved the analysis and structural identification of the steroidal glycosides in Polygonatum plants, by using an LC-ESI-MS method but without the help of high-resolution measurements. The molecular weight can be estimated by observation of characteristic molecular ions in positive and negative mode ESI-MS spectra, and the MS fragment patterns are also reasonably correlated with chemical structures, as described in the manuscript. Moreover, the HPLC behavior of each compound is in good agreement with the results. Taking everything into consideration, we believe that we have presented a correct and reliable result for this study. We would appreciate it if any mistake in the result could be pointed out, then we are pleased to address it.

Reviewer 3 Report

After appropriate amendments made by authors I recommend the manuscript for the publication in its present form.

Author Response

Thank you very much.